# Importance of Breed, Parity and Sow Colostrum Components on Litter Performance and Health

**DOI:** 10.3390/ani12101230

**Published:** 2022-05-10

**Authors:** Laura Amatucci, Diana Luise, Federico Correa, Paolo Bosi, Paolo Trevisi

**Affiliations:** Department of Agricultural and Food Sciences (DISTAL), Alma Mater Studiorum-University of Bologna, Viale G. Fanin 46, 40127 Bologna, Italy; laura.amatucci2@unibo.it (L.A.); federico.correa2@unibo.it (F.C.); paolo.bosi@unibo.it (P.B.); paolo.trevisi@unibo.it (P.T.)

**Keywords:** somatic cells, IgA, diarrhoea, piglet survival

## Abstract

**Simple Summary:**

Colostrum quality and quantity can influence the growth, health, and survival of sow offspring, and can therefore influence sow productive performance. Colostrum quality can be affected by a multitude of factors; of these, the breed of the sows and its parity were investigated in the present study. The aim of the study was to identify the influence of sow breed and parity on colostrum components and to associate these with the survival, growth, and occurrence of diarrhoea of their litters. The results revealed that a more robust breed, such as Duroc, could have more colostral immunoglobulins (Igs) than the Large White and Landrace breeds. Gilts are characterised by a higher fat % and number of somatic cells (SCs), which are positively correlated to each other. Piglet survivability until weaning can increase with an increased quantity of IgA and SCs in the colostrum; piglet diarrhoea can decrease with an increased quantity of IgA. Increasing knowledge regarding swine colostrum composition and its relation to litter performance can help in designing new intervention strategies to improve the welfare and economic sustainability of pig rearing systems.

**Abstract:**

The aims of this study were to investigate the effect of breed and parity on colostrum components, and to associate sow breed, parity, and colostrum components with survival, growth, and the occurrence of diarrhoea of their litters. In Experiment 1, 64 sows (Duroc = 13; Landrace = 17 and Large White = 34) were included. In Experiment 2, 71 sows with different parities (1 = 10; 2 = 16; 3 = 13; 4 = 12; ≥5 = 20) were included. The number (N) of live piglets, litter body weight (Experiment 1), and the occurrence of diarrhoea (Experiment 1) were recorded at farrowing, at 2–3 days of age, and at weaning. Colostrum was analysed for proximate composition, immunoglobulins (Igs), and somatic cell count (SCC). Stepwise regressions and ANOVA models were used to associate breed, parity, and colostrum components with litter performance. The Duroc breed had the highest IgG and IgA (*p* < 0.005). Gilts had a higher fat% and SCC (*p*< 0.0001); these compounds were positively correlated (r = 0.45). Increased IgA tended to increase the N of weaned piglets (*p* = 0.058) and reduce litter diarrhoea (*p* = 0.021). The SCC increased the N of weaned piglets (*p* = 0.031). Overall, this study confirmed that breed and parity can influence the colostrum composition and highlighted the key role of Igs and somatic cells in piglet health.

## 1. Introduction

Colostrum contains several compounds, such as lipids, proteins, amino acids, oligosaccharides, lactose, immunoglobulins (Igs), and vitamins, which are essential for the growth of newborns. In fact, good colostrum intake has been associated with an improvement in daily weight gain until weaning, an improvement in the body weight (BW) of weaned and fattening pigs, and reduced mortality during the suckling and nursery periods [1,2,3]. Moreover, in pigs, maternal immune cells and Igs cannot pass from mother to foetus during gestation because the placenta of sows is epitheliochorial. For this reason, it is very important that piglets acquire enough colostrum before their functional gut closure, which occurs 24–36 h after birth, to allow for Igs to pass from the colostrum to the piglet’s blood [4,5]. It has been demonstrated that Igs (IgG and IgA) are in higher concentrations in the colostrum than in the serum of sows, and that the piglet’s absorption of colostral Igs is nonselective; in fact, post-colostral piglets have a distribution of Igs in serum similar to that seen in sow colostrum [6]. Absorption of the Igs of colostrum by piglets is necessary to grant them passive immune protection; however, it also influences the amount of IgG which is actively produced by the piglets, the development of which starts at 7 days of age [7]. Therefore, it is necessary that piglets take at least a minimum amount of colostrum to guarantee their survival, growth, and development of immunity [8]. However, at the same time, it is equally important that colostrum is rich in Igs, the production of which varies according to the health status of the sow.

In other domestic species, the mother’s health status has been associated with the somatic cell count (SCC), which has been recognized as an indicator of milk quality [9]. The SCC is also an indicator of udder health and could in turn affect Igs secretion [9]. 

For dairy cows, in which colostrum and milk composition have seen more study in comparison to sows, the factors affecting the Igs concentration—the SCC and the colostrum composition—are essential for good health, calving season, breed, age, their production system, a healthy diet, length of non-lactating period, and the time delay between parturition and first milking [9]. Similar to cows, the composition of sow colostrum can be influenced by several factors, including those of the sow-independent variety, such as environmental conditions (including the temperature and quality of hygiene of the farm), the sow’s diet, and sow-dependent factors including breed, parity, and general health. For example, it has been observed that sow diet can influence their colostrum proximal composition and Igs concentrations [10,11]. Several studies have reported that genotype or breed can influence the composition of colostrum in terms of proteins, lactose, lipids, oligosaccharides, or metabolites [3,12,13,14]. Since sow breed can influence colostrum composition, as mentioned above, it can thus be hypothesised that it can also influence the quantity of Igs; however, currently, there are few data regarding this [15,16,17]. Data in the literature has suggested that colostrum quality and quantity can be greatly affected by sow parity. In fact, it is known that multiparous sows can produce more colostrum and milk because they do not need to convert energy for their own growth and maturation, unlike gilts [4]. It is also known that parity can influence the concentration of lipids, short-chain fatty acids (including total saturated fatty acids [SFA], total unsaturated fatty acids [UFA], total monounsaturated fatty acids [MUFA], and total polyunsaturated fatty acids [PUFA]), lactose, Igs concentration, and small metabolites (including *O*-acetylcholine, *O*-phosphocholine, and sn-glycero-3-phosphocholine) [3,18,19]. The effect of parity on Igs content in the colostrum could be due to the antigenic pressure on the sows, their health status, and their maturation. The colostral IgG and a high proportion of IgM in the colostrum and milk of sows are derived from the blood serum [20]. The same could be said for IgA, of which approximately 60% is produced at the mammary level [20], upon homing of the gut-derived plasma cells, particularly activated at the level of the Payer’s patches [21]. Therefore, the functional efficiency of the udder could be a conditioning factor for Igs release and may be connected to the SCC [9]. 

Although parity and breed are recognised as important factors affecting colostrum quality, little is yet known about their effects on colostrum proximal composition, Igs concentration, and the SCC in swine colostrum, as well as their consequences in the litters. Therefore, the present study had the aim of investigating the effects of different Italian breeds (Italian Large White [LW], Italian Landrace [L], and Italian Duroc [D]) and sow parity on proximate composition, Igs concentration, and somatic cells (SCs) quantity in the colostrum to assess the relationship among these components, as well as to disclose the most important features affecting piglet growth and survivability.

## 2. Materials and Methods

The procedures complied with Italian law pertaining to experimental animals and were approved by the Ethic–Scientific Committee for Experiments on Animals of the University of Bologna, Italy. 

To elucidate the effect of Italian breeds and sow parity on colostrum composition, Igs concentration, and the SCC and their relation with the litter, two independent experiments were carried out.

### 2.1. Experiment 1

The aims of Experiment 1 were to do the following: (1) assess the effect of the Italian breed on Igs concentration in the colostrum; (2) test the associations between sow breed, colostrum Igs concentration, survival, growth, and diarrhoea occurrence in piglets. 

For Experiment 1, colostrum samples were collected from 64 sows raised on the same farm from May to August of the following year. The sows belonged to the three different breeds which are the most important in Italian pig production: Italian Duroc (13), Italian Landrace (17), and Italian Large White (34). All the sows were raised indoors under the same environmental conditions, with an automated system to regulate temperature and humidity; the European Union (EU) rules for guaranteeing pig welfare were followed. Four weeks after insemination, the sows were kept in groups of 10 and, five days before farrowing, they were moved into the farrowing room into single cages. All the sows had free access to water for the experimental period. All animals were fed twice a day with 2.5 kg of the same commercial diet comprised, in decreasing order, of barley (42%), wheat bran (30%), wheat flour (11%), soybean meal (7%), corn (4.30%), whole soybean (2%), fish oil (0.50), sodium chloride (0.4%), mycotoxin binder (0.2%), L-lysine monohydrochloride (0.15%), choline (0.11%), magnesium sulphate anhydrous (0.05%), threonine (0.05%), and methionine (0.04%). The diet resulted in the following composition: crude protein (16.48%), crude fat (3.70%), crude fiber (7.27%), starch (37.57), and starch + sugar (41.03%), with a digestible energy of 6641.52 kcal/d.

For the study, only healthy animals which were not treated with antibiotics or other drugs during gestation and lactation were considered. Farrowing was not induced, and the colostrum was collected across all teats of the sows in the period between the birth of the first piglet and before the birth of the last [3]. Sows which needed farrowing induction or had long parturitions were excluded from the study to avoid confounding factors on the colostrum composition. After collection, the samples were immediately frozen in liquid nitrogen and then stored at −80 °C.

Parity, date of farrowing, and productive performance data were recorded for each sow. The N (N) of live piglets and the litter body weight (LBW) were recorded at birth, at 3 days of age (d3), and at weaning (d25). The weight of the dead piglets was removed from the LBW. Furthermore, the occurrence of diarrhoea during suckling (1 = presence of diarrhoea events from piglet birth until weaning; 0 = absence of diarrhoea events) was recorded.

### 2.2. Experiment 2

The aims of Experiment 2 were as follows: (1) to assess the correlation among the macronutrient composition, Igs concentration, and the SCC in the colostrum; (2) to evaluate the effect of parity on the colostrum components; (3) to test the associations between parity, the colostrum components, and the productive performance of the sows up to weaning. 

A total of 71 sows (PIC hybrid line; parity 3.74 ± 2.02; litter size= 14.46 ± 2.77) were included in Experiment 2. Following the European regulations, upon entering the delivery room 5 days before farrowing, the sows were housed in single farrowing crates of 4.5 m^2^. All the sows were raised indoors under the same environmental conditions and were fed the same standard lactation diet based on corn (20.68%), wheat bran (20.0%), barley (14%), wheat (12%), sorghum (8%), dried sugar beet pulp (6.5%), dried brewers’ grains (5%), soya bean meal with 48% crude protein (4%), sunflower meal with 36% crude protein (3%), and whole linseed (2.5%). The diet resulted in the following composition: crude protein (14.2%), fat (4.4%), crude fiber (5.99%), ash (5%), starch (36.7%), sugars (3%), neutral detergent fibre [NDF] (20.1%), fibre [F] (20.1%), and acid detergent fibre [ADF] (8%) with a metabolisable energy of 2932 kcal/kg. The sows were fed using an automatic system, according to the feeding plan normally used on the farm, for 8 days during the peripartum period, with an average of 5 days before delivery and 3 days post-partum (including the day of delivery): 2.5 kg feed per day, 1 kg on the day of farrowing, 1.5 kg feed on the first day of lactation, 2.5 kg on the second, and then ad libitum. Colostrum samples were collected at farrowing in the same manner as in Experiment 1. 

The N of live, dead, and total piglets was recorded for each sow at farrowing, after cross-fostering, at 2 d of age, and at weaning 23–25 d of age. Cross-fostering of piglets was performed within one day post-farrowing to balance the N of suckled piglets per sow. After cross-fostering, the dead pigs were never replaced by additional cross-fostering piglets. The individual body weight (BW) of the piglets at birth and at weaning were recorded.

### 2.3. Colostrum Analysis

The colostrum samples retained from Experiment 1 and Experiment 2 were analysed for Igs concentration. The Igs concentration (namely, IgA, IgM, and IgG) in the colostrum was analysed using an immunoglobulin enzyme-linked immunosorbent assay (ELISA) protocol according to [11]. The reaction was quantified spectrophotometrically at an absorbance of 405 nm using a microplate reader (Multiskan FC Microplate Photometer—Thermo Fisher Scientific). For the analysis, the colostrum samples were diluted at 1:50,000, 1:10,000, and 1:500,000 for IgA, IgM, and IgG, respectively. The detection limits were 21.4–1300 ng/mL for IgA, 15.6–1000 ng/mL for IgM, and 7.8–1000 ng/mL for IgG. Of note, the intra- and inter-assay coefficients of variation (CVs) for these ELISA assays were between 3 and 25%, respectively. Concentration values, expressed in mg/mL, were calculated using a four-point parametric curve.

Furthermore, the colostrum samples of Experiment 2 were analysed for their proximate composition and SCC. The composition of sow colostrum was analysed in triplicate for protein, fat, lactose, and urea content, and SCC with infrared spectroscopy using a Milkoscan FT2 (FOSS A/S, Padova, Italia).

### 2.4. Statistical Analysis

The first objective of the study was to investigate the effect of breed and parity on the colostrum components; therefore, ANOVA models were built to test this assumption in the two independent experiments. The second objective was to associate sow performance at farrowing with sow performance during the suckling period, together with breed, parity, and colostrum composition. Therefore, the data were initially analysed using Stepwise Regression analysis. The factors significantly associated with sow performance during the suckling period, obtained from the Stepwise Regression analysis, were then included in an ANOVA model to confirm their significance. 

#### 2.4.1. Experiment 1

Data regarding Igs concentration and sow performance at birth were analysed using an ANOVA model, including breed and parity (from 1 to 4: 1 = 6; 2 = 17; 3 = 22; 4 = 19) order as fixed factors. Season (from 1 to 4) and N of live piglets at birth were initially included and then removed, as they were deemed insignificant. A Tukey’s honest significance test was then carried out at a 95% confidence level (*p* < 0.05). A Stepwise Regression analysis was then used to select, among the variables, those influencing the N of dead piglets from birth until d3, the N of piglets which died from farrowing to weaning, the LBW at weaning, the average daily gain (ADG) of the piglets from birth to weaning, and the occurrence of diarrhoea from birth to weaning. The significant factors obtained from the stepwise analysis were used to build ANOVA models to associate the colostrum components, the breed and the sow performance at farrowing to growth, survival, and diarrhoea of piglets until weaning. Values were considered to be significant when *p* was <0.05, and to be a tendency when *p* was ≤0.10.

#### 2.4.2. Experiment 2

The Pearson correlation was carried out among the colostrum components using the “Hmisc” package in R software. The correlations were then visualised using the “cor-plot” package implemented within the R environment. Sow performance and colostrum components at farrowing were analysed using an ANOVA model including parity (1 = 1° 2= 2°, 3 = 3° 4 = 4° and 5° and 5 > 5°) and several covariates, depending on the parameters investigated. Those included in the final models are reported in the Tables. A stepwise regression analysis including colostrum components and sow productive performance at farrowing was carried out to select the variables which influenced the percentages of dead piglets at 24 h, 2 d, and at weaning, and the piglets’ ADG. The ANOVA models were then built to associate the colostrum components, parity class, and the factors which were significant from the stepwise analysis with the percentage of dead piglets at 24 h, 2 d, and at weaning, and the piglets’ ADG. Values were considered to be significant when *p* was <0.05, and to be a tendency when *p* was ≤0.10.

## 3. Results

### 3.1. Experiement 1

Table 1 shows the effect of breed and parity on sow performance at birth. Breed significantly influenced the N of piglets born live (*p* = 0.002). The N of piglets born live from the L and the LW sows was significantly higher compared to the D sows (*p* = 0.001 and *p* = 0.02, respectively). No difference between the L and the LW sows was found concerning the N of piglets born live. Breed significantly influenced the LBW at birth (*p* = 0.022). In the L breed, the LBW at birth was higher than in the D breed (*p* = 0.012), and there was a tendency to differ between the LW and the D breeds (*p* = 0.06). Regarding piglet BW at birth, there were no significant differences between the three breeds; only a tendency for a higher BW in the D breed compared to the L breed (*p* = 0.10) was observed. The N of stillborn piglets was not affected by the breed. The parity of the sows affected the LBW at birth (*p* = 0.055;1 = 13.1;2 = 15.1;3 = 16.7;4 = 16.6) and tended to influence the N of stillborn piglets (*p* = 0.069; 1 = 0.77; 2 = 0.69;3 = 1.15;4 = 0.38).

Figure 1 shows the effect of breed on the Igs concentration in the colostrum. The IgA concentration was significantly influenced by the sow’s breed (*p* = 0.046). The IgA concentration was higher in the D breed compared to the LW breed (*p* = 0.029) and tended to be higher in the D breed compared to the L breed (*p* = 0.089). The IgM concentration was not influenced by the breed. The IgG concentration was significantly influenced by the breed (*p* = 0.004). The IgG concentration was higher in the D breed compared to the LW breed (*p* = 0.002) and tended to be higher in the L breed compared to the LW breed (*p* = 0.09). No differences between the LW and the L breeds were found for IgA and IgM concentrations.

Stepwise Regression and the subsequent ANOVA analysis revealed that breed, sow productive performance (N of piglets born live and BW of piglets at birth), and concentrations of Igs were associated with piglet survival, growth parameters, and the occurrence of diarrhoea (Table 2). 

The LBW at weaning and the N of piglets at weaning were influenced by the breed (LBW at weaning: *p* = 0.001; D = 46.97; L = 71.12; LW = 71.95; N of piglets at weaning: *p* = 0.014; D = 7.66; L = 10.32; LW = 9.67); in addition, the N of piglets at weaning tended to increase with the concentration of colostrum IgA (*p* = 0.058; coeff = 0.043). The presence of diarrhoea in the litter was reduced by the increasing IgA concentration in the colostrum (*p* =0.021; coeff= −0.0089) and was influenced by the breed (*p* = 0.025; D = 0.36; L = 0.00; LW = 0.10).

### 3.2. Experiement 2

Table 3 shows the effect of sow parity on sow performance at farrowing. The parity significantly affected the piglets’ BW at birth (*p* = 0.02) and tended to have linear (*p* = 0.091) and quadratic (*p* = 0.096) effects; the gilts had lighter piglets than sows of second parity (*p* = 0.020). The N of live piglets at birth tended to be influenced by parity (*p* = 0.057), which itself tended to have a linear (*p* = 0.059) effect; the gilts had a higher N of live piglets at birth, compared to the sows having ≥5 parities. Piglet birth BW was reduced by the N of live piglets (*p* = 0.028; coef= −21.53) and, conversely, the N of live piglets at birth was reduced by the piglet BW at birth (*p* = 0.028; coef= −0.003). The N of dead piglets at birth was reduced with a higher piglet BW (*p* = 0.002; coef = 0.002). The N of dead piglets at birth and the N of piglets after cross-fostering were not influenced by parity.

Table 4 shows the effect of parity on composition, SCC, and Igs of the colostrum. The percentages of protein, casein, urea, and Igs were not significantly influenced by the parity of the sows. The percentage of fat was significantly affected by parity (*p* < 0.0001), and gilts had a higher fat percentage in comparison to the others’ parities. Moreover, the parity showed linear (*p* < 0.0001) and quadratic (*p* = 0.001) effects on fat percentage. The N of SCs was significantly affected by parity (*p* < 0.0001); it showed linear (*p* < 0.0001) and quadratic (*p* < 0.001) effects and was higher in gilts in comparison to the other parities (*p* < 0.0001). The percentage of lactose was significantly lower in parity 2 sows than in sows of parity >5 (*p* = 0.01). Moreover, the parity tended to have a linear effect on the percentage of lactose (*p* = 0.10).

Figure 2 shows the correlations among colostrum components. There were high negative correlations between IgA and lactose (*p* < 0.0001; r = −0.51), between protein and lactose (*p* < 0.0001; r = 0.64), and casein and lactose (*p* < 0.0001; r = −0.64). Furthermore, negative correlations were observed between IgA and IgM (*p* = 0.009; r = −0.31), lactose and IgG (*p* = 0.018; r = −0.28), and between lactose and fat (*p* = 0.02; r = 0.27). On the contrary, there were positive correlations between fat and the N of somatic cells (*p* < 0.0001; r = 0.45), fat and IgM (*p* = 0.036; r = 0.25), protein and IgA (*p* = 0.004; r = 0.39), and casein and IgA (*p* = 0.004; r = 0.39). 

The stepwise regression analysis revealed that, in addition to the litter characteristics, specific colostrum components were influencing the percentage of dead piglets at 24 h, and the percentage of weaned piglets while the percentage of dead piglets 2 d post-farrowing and the LBW at weaning were not influenced by any colostrum components. Results of the ANOVA model including the significant factors detected using the stepwise regression are reported in Table 5. The percentage of fat and the concentration of IgA entered into the stepwise model regarding the percentage of dead piglets at 24 h but was not significant after the ANOVA model (*p* > 0.1). The percentage of dead piglets from 24 h to weaning and the percentage of total dead piglets were not affected by the colostrum components. The percentage of weaned piglets increased with s increase of SCC (*p* = 0.033; coef = 0.007). The LBW tended to decrease, with a higher concentration of IgM in the colostrum (*p* = 0.07; coef = −146). The litter ADG decreased with a higher concentration of IgM in the colostrum (*p* = 0.029; coef = −8.51).

## 4. Discussion

This study confirmed that the components of sow colostrum can be influenced by different factors, including the breed and parity of the sows, and that specific components could help explain sow performance in terms of growth and survivability of their litters. 

It is widely known that breed can greatly influence sow performance at birth [22,23]. The results of the present study have confirmed the effect of breed on sow performance at birth, as the L and LW sows showed a greater N of piglets born live and increased litter BW at birth than the D sows. The present study agreed with the our previous study in which L and LW sows had larger litters at birth than D sows [13]. Also in the present study, the piglet BW at birth did not change among these three breeds; on the contrary, Knecht et al. (2015) [24], whose study focuses on Polish Landrace, Polish Large White, and Polish Landrace x Polish Large White crossbreeds, and Quesnel et al. (2008) [22], sustained that breed can also influence piglet birth and weaned weight. In addition to litter performance, breed is known to influence some components of swine colostrum, including fat, lactose, and oligo-saccharide composition [3,13,14,25,26]. In the present study, of the components analysed in the colostrum, breed influenced Igs concentration; in the D sow colostrum, there was a higher concentration of IgA and IgG than in the LW sows’ colostrum. There are only a few studies which show how Igs concentration (IgA and IgM) in the colostrum is influenced by the breed of the sow [15,16,17]. Duroc is considered a robust breed characterised by a lower N of piglets born, and more concentrated and fattier colostrum in comparison to the LW and the L breeds [3,13]. The fact that the colostrum and the milk from the Duroc breed is more concentrated may also contribute to explaining the higher Igs concentration observed in that breed in comparison to the LW breed.

Parity is another known factor which can affect swine colostrum composition, as has previously been reported in several studies [27,28,29]. In the present study, parity significantly influenced the fat and lactose concentration, as well as the SCC. In agreement with the present study, other studies have reported that the concentration of fat in the colostrum is influenced, or tends to be influenced, by the parity of the sow, with a higher concentration in primiparous sows [28,30]. The effect of parity on lactose concentration is more controversial; in contrast with the present study, other studies have reported that the concentration of lactose in the colostrum is not influenced by parity [4,28,30,31]. However, lactose concentration in the milk of sows is influenced by parity, as has been reported by Beyer et al. 2007 [27]. The effect of parity on fat and lactose concentrations may also be influenced by the inverse correlation between these two components, as observed in the present study. Of the colostrum components, the fat content of mammary secretions is one of the most variable of all components [31]. In fact, the colostrum fat percentage increases with the reduction of colostrum yield [31,32]. Conversely, lactose concentration is positively associated with colostrum yield [32]. This could explain why fat is more concentrated in primiparous sows than in multiparous sows since sows of fourth parity have the highest productivity of milk, which is also higher in lactose [27]. The present study also showed that the SCC is higher in gilts, and it decreases with the increase in parity. The greater N of SCs observed in primiparous sows agreed with some studies carried out on cows in which it is known that the SCC is higher in younger cows [33,34]. The SCC consists of leukocytes and epithelial cells; the present results would suggest that the mammary gland of young animals may have a different sanitary or maturation status than mature animals. The different status of the maternal gland of young animals was also confirmed by a trend to a lower IgA concentration. Carney–Hinkle et al. (2013) [35] showed that IgA concentration tended to be greater in fourth parity sows than in primiparous sows. Regarding the other Igs, namely IgM and IgG, in agreement with the present study, their concentration did not seem to be affected by the parity of the sow [18,29,35].

The present study showed that the litter ADG was higher in multiparous sows than in gilts, and especially in sows of parity 4. This could have been due to the differences in the colostrum composition observed in the present study; in addition, as sustained by different studies, the colostrum and milk yield in primiparous sows is lower than in multiparous sows [18,36]. In fact, primiparous sows are probably unable to provide prolonged lactation, which is due to a drop in lactose in the late phases of lactation, as compared to multiparous sows [4].

Interestingly, in the present study, it was observed that proximal components, Igs concentration, and SCC could be correlated. The IgA concentration was negatively correlated with IgM concentration. This could have been due to the fact that these two Igs have many similarities in their structure; their polymeric structure allows them to be delivered to the mucosal epithelium [37], and they constitute the passive mucosal protection of neonatal piglets [38]. Therefore, it could be possible that there is a competition in the release of IgA and IgM due to the switch from IgM to IgA as a secondary immune response [39]. Furthermore, a positive correlation was observed between the SCC and fat, as previously observed by Maurer et al. (2020) [40]. At the same time, the study showed that lactose was negatively correlated with protein, casein, IgA, IgG, and fat. Lactose is the most important osmotic compound in mammary secretions; its production is important to transfer water to the alveoli, and thus for the volume of the colostrum [19,32]. The increase of lactose probably induces a higher recall of water, and the concentration of other components decreases. To sustain this hypothesis, Hurley (2015) [31] reported that, on the first postpartum day, there is an increase in water content in the colostrum which results in a decrease in proteins—mainly Igs—and an increase in lactose.

The present study has also attempted to investigate a possible relationship between colostrum components and litter survivability and growth until weaning. In fact, it is known that colostrum intake within the first 24 h of life and colostrum Igs concentration can significantly affect the survivability of neonatal piglets, as Igs plays a significant role in the development of the mucosal (IgA) and systemic immunity (IgG) of piglets [18,41]. As previously reported, in this study, the N of piglets weaned was associated with the N of piglets in the litter [42,43], with the BW of the piglets, and with the sow breed. However, colostrum components—namely, the concentration of IgA and the N of SCs—could contribute to increasing the N of weaned piglets. Babicz et al. (2011) [44] reported that gilts with lower SCC values had higher milk yields and litters with a higher BW and daily weight gain. On the contrary, Maurer et al. (2020) [40] observed that SCs do not influence colostrum composition—except for the percentage of lipids—or litter performance. Furthermore, in the present study, it was observed that, in addition to the N of weaned piglets, the IgA concentration could significantly reduce diarrhoea in the litter. This result agreed with a study by Hasan et al. (2019) [45], who reported that the incidence of diarrhoea in piglets was greater when the level of IgA in the colostrum was lower, but also when the level of serum amyloid A was lower, or when sow serum progesterone was higher. Studies have shown that colostrum IgA was correlated with intestinal mucosal immunity, reporting the importance of high IgA levels in milk to provide local protection for suckling piglet intestinal tracts and to protect them from porcine epidemic diarrhoea virus (PEDV) [46,47]. Finally, it was observed that the concentration of colostrum IgM was negatively correlated with litter BW and litter ADG. This could have been due to the negative correlation between IgM and IgA; in fact, the IgA was positively correlated with litter BW. 

The present study confirmed the importance of piglet BW at birth as regards LBW and the survivability of piglets during the suckling period [1,36,43,48]. Furthermore, it is important to highlight that individual feed intake of the sows and their body condition score, which are closely linked to productivity (litter performance) and the quality and quantity of colostrum [49], were not recorded in the present study. Therefore, additional studies including this information are advisable to unravel the relationship between sows, the colostrum and milk composition, and piglet growth and survivability. 

## 5. Conclusions

Overall, the aforementioned results improved knowledge regarding swine colostrum composition and the effect of breed and parity on it. In particular, the study provided insights into the role of SCs in the swine colostrum. Additional research regarding the relationship between piglet survival and growth performance, as well as the colostrum components, is advisable to edify management and nutrition strategies aimed at providing good-quality colostrum to piglets.

## Figures and Tables

**Figure 1 animals-12-01230-f001:**
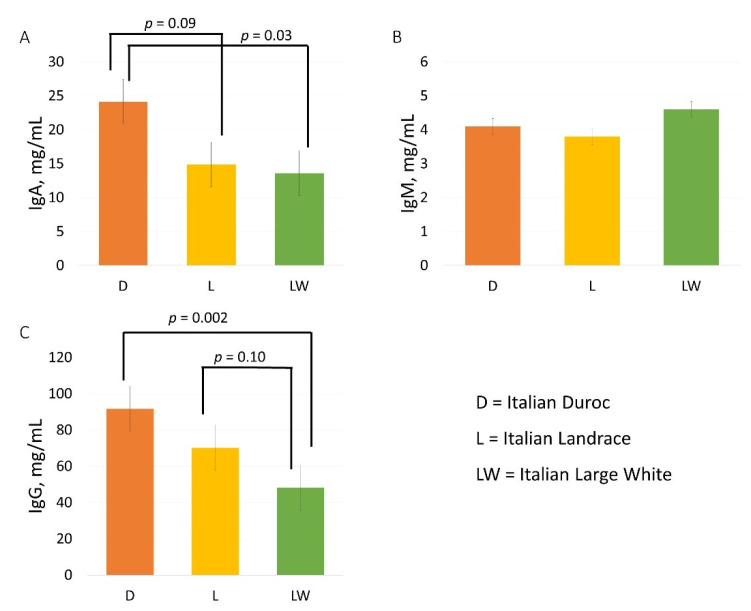
**The effect of breed on the concentration of immunoglobulins in swine colostrum (least square means with SEM).** (**A**) = Effect of breed on the concentration of IgA; (**B**) = Effect of breed on the concentration of IgM; (**C**) = Effect of breed on the concentration of IgM.

**Figure 2 animals-12-01230-f002:**
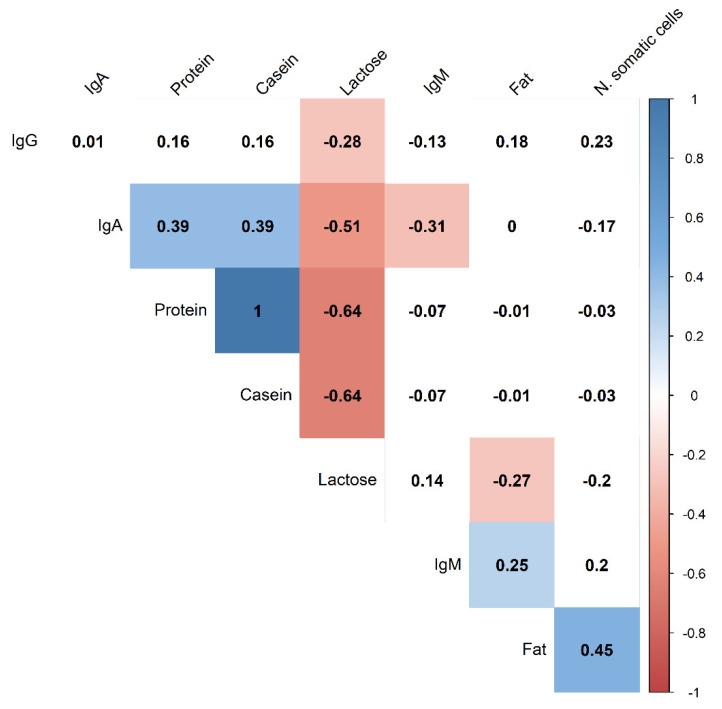
**Pearson correlation among the components of sow colostrum.** Red identifies negative significant (*p* < 0.05) correlations. Blue identifies positive significant (*p* < 0.05) correlations. White identifies no significant correlations.

**Table 1 animals-12-01230-t001:** The effect of breed and parity on sow performance at birth.

Performance	Mean				*p*-Value				
	D	L	LW	SEM	Breed	Parity	D vs. L	D vs. LW	L vs. LW
N of piglets born live	9.00	12.77	11.59	0.538	0.002	0.412	0.001	0.020	0.229
LBW at birth, kg	14.03	17.40	16.43	0.737	0.022	0.055	0.012	0.060	0.509
Piglet BW at birth, kg	1.59	1.38	1.40	0.052	0.121	0.965	0.101	0.125	0.917
N of stillborn piglets	1.08	0.71	0.59	0.235	0.602	0.069	0.837	0.520	0.795

D: Italian Duroc; L: Italian Landrace; LW: Italian Large White; SEM: standard error of mean. Parity = parity order.

**Table 2 animals-12-01230-t002:** Results of the ANOVA analysis on sow performance at three days post farrowing and at weaning.

Variable	Coefficient	SE Coefficient	Mean	SE Mean	T-Value	*p*-Value
**Model for LBW at d3**						
N of piglets born live	1.280	0.158			8.17	<0.0001
BW of piglets at birth	8.310	2.24			3.7	0.001
**Model for LBW at weaning**						
N of piglets born live	1.895	0.844			2.25	0.031
Colostrum IgA (mg/mL)	0.272	0.175			1.55	0.129
Breed						0.001
D			46.97	5.19		
L			71.12	4.19		
LW			71.95	3.08		
**Model for N of piglets weaned**						
N of piglets born alive	0.366	0.104			3.52	0.001
Colostrum IgA (mg/mL)	0.043	0.022			1.94	0.058
Breed ^1^						0.014
D			7.67	0.66		
L			10.33	0.518		
LW			9.67	0.349		
**Model for diarrhoea in the litter**						
Colostrum IgA (mg/mL)	−0.009	0.004			−2.39	0.021
Breed ^1^						0.025
D			0.36	0.102		
L			−0.02	0.085		
LW			0.11	0.059		

Breed ^1^ D = Italian Duroc; L = Italian Landrace; LW = Italian Large White.

**Table 3 animals-12-01230-t003:** Effect of sow parity on sow performance at farrowing.

Item	Parity, Mean					SEM	*p*-Value				
	1	2	3	4	≥5		Parity	Linear	Quadratic	N of Live Piglets	Piglet Birth BW
N. of sows	10	16	13	12	20						
Piglet birth BW, g	1162 ^a^	1449 ^b^	1310 ^ab^	1351 ^ab^	1370 ^ab^	56	0.020	0.091	0.096	0.028; coef = −21.53	-
N. of live piglets at birth	15.1 ^a^	15.2 ^ab^	14.8 ^ab^	15.1 ^ab^	13 ^b^	0.7	0.057	0.059	0.15	-	0.028; coef = −0.003
N. of dead piglets at birth	0.57	0.88	0.51	0.89	1.14	0.3	0.591	0.245	0.632	0.902	0.002; coef = −0.002
N. of piglets post cross-fostering	13.6	13	13.8	12.8	13.2	0.42	0.423	0.46	0.78	-	-

^a^, ^b^: Values differ at *p* < 0.05.

**Table 4 animals-12-01230-t004:** Effect of sow parity on the colostrum composition, the number of somatic cells, and the immunoglobulins.

Item	Parity, Mean					SEM	*p*-Value			
	1	2	3	4	>5		Parity	Partition of Parity Effect		N of Live Piglets
								Linear	Quadratic	
Fat, m/m	8.91 ^a^	6.24 ^b^	6.27 ^b^	5.25 ^b^	6.19 ^b^	0.42	<0.0001	<0.0001	0.001	0.601
Protein, m/m	22.8	23.9	23.2	22.9	22.9	0.5	0.444	0.616	0.331	0.259
Casein, m/m	5.96	6.61	6.19	6.02	6.01	0.25	0.444	0.616	0.331	0.444
N. of somatic cells, n/1000 mL	8524 ^a^	3039 ^b^	2826 ^b^	1246 ^b^	1256 ^b^	731.8	<0.0001	<0.0001	0.001	0.942
Lactose, m/m	3 ^ab^	2.7 ^a^	3.11 ^ab^	3.21 ^b^	3.05 ^ab^	0.11	0.016	0.108	0.979	0.735
Urea, mg/100 mL	50.7	52	51.1	51.9	52.7	1.76	0.928	0.471	0.929	0.955
IgM, mg/mL	2.81	2.21	2.09	2.49	1.99	0.25	0.132	0.137	0.461	0.903
IgG, mg/mL	91.7	58	38.9	26.3	72.1	21	0.242	0.3	0.037	0.099
IgA, mg/mL	10.9 ^b^	16.6 ^a^	14.4 ^ab^	13.3 ^ab^	16.2 ^ab^	1.8	0.09	0.2	0.53	0.07

^a^, ^b^: Values differ at *p* < 0.05.

**Table 5 animals-12-01230-t005:** Results of the ANOVA analysis regarding sow performance at 24 h, 2 days post-farrowing, and weaning.

Variable.	Coefficient	SE Coefficient	*p*-Value
**Model for % of dead piglets at 24 h**			
Fat % colostrum	0.76	0.49	0.13
IgA, mg/mL colostrum	−0.26	0.17	0.13
Post cross-fostering BW of the piglets	−2.98	0.66	<0.0001
N. piglets post cross-fostering	−0.01	0.01	0.08
**Model for % of dead piglets from 24 h to weaning**			
SCC, n/1000 mL colostrum	−0.0004	0.0001	0.13
Post cross-fostering BW of piglets	−0.019	0.005	<0.0001
**Model for % of total dead piglets**			
IgA, mg/mL colostrum	−0.34	0.25	0.18
N of post cross-fostering piglets	−1.51	1.01	0.14
Post cross-fostering BW of piglets	−0.02	0.01	0.01
**Model for % of weaned piglets**			
SCC, n/1000 mL colostrum	0.007	0.0003	0.03
N of post cross-fostering piglets	−1.76	0.75	0.02
Post cross-fostering BW of piglets	0.02	0.01	0.001
**Model for LBW**			
Post cross-fostering BW of piglets	1.93	79.6	<0.0001
IgM, mg/mL colostrum	−146	0.36	0.07
**Model for Litter ADG**			
IgM, mg/mL colostrum	−8.51	3.8	0.03
N of post cross-fostering piglets	−5.07	2.15	0.02
Parity	^A^	*	0.03

^A^ Least Squares means: 1 = 203; 2 = 233; 3 = 219; 4 =235; ≥5 =220. * Parity 1 vs. Parity 4, *p* = 0.051.

## Data Availability

The data can be found within the article.

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
