# Peer review of "Importance of Breed, Parity and Sow Colostrum Components on Litter Performance and Health"

_animals, 2022, doi:10.3390/ani12101230_

Round 1

Reviewer 1 Report

Review Report

  • A summary

The study was aimed at investigating the effects of breed and parity on the colostrum composition and quality. The approach was based on two in-vivo experiments. The first experiment assessed the effect of Italian breeds on the concentration of Igs in colostrum. Correlation between breed colostrum Igs and survival, growth performance and diarrhoea occurrence of the offspring were assessed. In the second experiment effect of parity on the colostrum composition and the association between parity and colostrum components and the sow’s productivity were studied.

  • Broad comments

The experiments seemed to be carefully done and the paper is thoroughly written. However, some reformulations are needed in the materials and methods and discussion parts. I would suggest that the results be discussed more systematically. Some unnecessary results are reported. The methods should be more adequately described (some important information about sow performance is missing), and the limitations should be mentioned in the discussion part. 

  • Specific comments

Page 1 line 27: change N to the number of.

Page 3 line 115: Mention composition of the diets and type of feeding strategy. Did you record the individual feed intake through the last pregnancy stage and during the lactation period?

Page 3 line 144: Did you record the individual feed intake through late pregnancy and during lactation period?

Page 4 line 177: Mention methods for test assumption.

Page 4 line 185: day birth à farrowing

Page 4 line 185: First use of ADG. It should be written completely.

Page 4 line 193: Did you include all the sows in Pearson correlation?

Page 6 line 235: I would suggest reformulating the figures. Igs (mg/ml) can stand at y-axis. The significance can be illustrated by legends (*) in the figure and not as a text under the figure. The abbreviation of groups can stand on the x-axis. Mark figures as subgroups (a-c) and describe information in the capture.

Page 6 line 241: Three days post post-birth farrowing

Page 7 lines 251-254: Reformulate the paragraph. Some results are unnecessary to be reported. For example, that LBW has been changed by number of the piglets and individual bodyweight of the piglets is not the most interesting finding.

Page 8 line 276: N of SCC

Page 10 line 355: Feed intake, growth and BCS of the sows are closely linked to productivity (litter performance) and the quality and quantity of colostrum. I do not see that these important parameters have been recorded in the experimental design. If these are not registered in the current study, they should be thoroughly discussed in the discussion section.

Author Response

  • Broad comments

The experiments seemed to be carefully done and the paper is thoroughly written. However, some reformulations are needed in the materials and methods and discussion parts. I would suggest that the results be discussed more systematically. Some unnecessary results are reported. The methods should be more adequately described (some important information about sow performance is missing), and the limitations should be mentioned in the discussion part. 

Authors: Thanks for your notable comments and suggestions. We revised the manuscript accordingly. Furthermore, the manuscript has been revised by a native English speaker to improve the language.

  • Specific comments

Page 1 line 27: change N to the number of.

Authors: Thanks, we revised it.

Page 3 line 115: Mention composition of the diets and type of feeding strategy. Did you record the individual feed intake through the last pregnancy stage and during the lactation period?

Authors: Thank you for the comment. We included additional information regarding the diets and feeding strategy. Unfortunately, it was impossible for us to measure individual feed intake because the in vivo trial had been performed on a commercial farm with a non-automated system. We agree with the reviewer that this information is important and that it could be useful. We will consider recording this information in future experiments. 

Page 3 line 144: Did you record the individual feed intake through late pregnancy and during lactation period?

Authors: Thank you for the comment. As mentioned before, unfortunately, it was impossible for us to measure individual feed intake because the in vivo trial had been performed on a commercial farm with a non-automated system. 

Page 4 line 177: Mention methods for test assumption.

Authors: Thank you for the comment. We revised the section including additional information; see lines 174-181.

Page 4 line 185: day birth à farrowing

Authors: Thank you for the correction. We revised the manuscript

Page 4 line 185: First use of ADG. It should be written completely.

Authors: Thank you for the correction. We revised the manuscript

Page 4 line 193: Did you include all the sows in Pearson correlation?

Authors: Thank you for the comment. We did not include the sows in the correlation matrix since we had one sample from each sow. 

Page 6 line 235: I would suggest reformulating the figures. Igs (mg/ml) can stand at y-axis. The significance can be illustrated by legends (*) in the figure and not as a text under the figure. The abbreviation of groups can stand on the x-axis. Mark figures as subgroups (a-c) and describe information in the capture.

Authors: Thank you for the comment. We revised the figure.

Page 6 line 241: Three days post post-birth farrowing

Authors: Thank you for the correction. We revised the manuscript.

Page 7 lines 251-254: Reformulate the paragraph. Some results are unnecessary to be reported. For example, that LBW has been changed by number of the piglets and individual bodyweight of the piglets is not the most interesting finding.

Authors: Thank you for the comment, we revised the paragraph of the results for both the experiments.

Page 8 line 276: N of SCC

Authors: Thank you for the correction.

Page 10 line 355: Feed intake, growth and BCS of the sows are closely linked to productivity (litter performance) and the quality and quantity of colostrum. I do not see that these important parameters have been recorded in the experimental design. If these are not registered in the current study, they should be thoroughly discussed in the discussion section.

Authors: Thank you for the comment, we added these considerations in the discussion.

Reviewer 2 Report

Comments to the Authors of manuscript number: animals-1696346 entitled “Importance of breed, parity and sow’s colostrum components on litter performance and health”.

The authors have presented a study on sows of various breed and parity, and their litters in relation to colostrum and milk composition. It is very interesting study, needed many time to perform. It fits to Animals.

  1. L L 17-18 – “positively correlated related’?
  2. L 32 – sows or gilts? There is no information if these females were primiparous or multiparous.
  3. L 40 amino acids
  4. L 66 – what the environmental condition’? does it relates to thermal condition? Explain it please
  5. L 75-76 – Is it truth? The mating is performed when females is mature physically, while sexual maturation appears earlier? Of course, the primiparous are smaller than sows after a few parities. Moreover, pigs do not have the finished growth which results from the construction of the growth plate. The rate of growth is slower only with time.
  6. L 77 fat? Lipid
  7. L 77 – saturated C15:0 or C16:0? MUFA or PUFA or OCFA
  8. L 77 what are these small metabolites? What do you mention about?
  9. L 113-114- energy
  10. Feeding description for two experiments should be the same.
  11. part of 2.3 the description of ELISA protocol is not needed. The detection limit and validation should be given
  12. L 263- it is not discovered for the first time
  13. L 370 – FA belong to fat
  14. L 379 – please explain why they have higher concentration of IgA
  15. L 399 – what is health status of mammary gland?
  16. in general the presence of SCC in the milk is not good. Mainly when these cells are leucocytes

Author Response

The authors have presented a study on sows of various breed and parity, and their litters in relation to colostrum and milk composition. It is very interesting study, needed many time to perform. It fits to Animals.

Authors: thanks for the comments and suggestions. We revised the manuscript accordingly to them and we hope it is now improved.

1.  L 17-18 – “positively correlated related’?

Authors: Thank you for the comment, we revised the sentence.

2. L 32 – sows or gilts? There is no information if these females were primiparous or multiparous.

Authors: Thank you for the comment. Gilts had had higher fat% and SCC than multiparous sows. Furthermore, the fat% and SCC had a positive correlation in colostrum considering all the available samples. We revised the sentence.

3. L 40 amino acids

Authors: Thank you for the comment, we revised the sentence.

4. L 66 – what the environmental condition’? does it relates to thermal condition? Explain it please

Authors: Thank you for the comment, we included additional information.

5. L 75-76 – Is it truth? The mating is performed when females is mature physically, while sexual maturation appears earlier? Of course, the primiparous are smaller than sows after a few parities. Moreover, pigs do not have the finished growth which results from the construction of the growth plate. The rate of growth is slower only with time.

Authors: Thanks for the comment. The mating is usually performed when the gilts have sexual maturation but this does not correspond to the complete growth of their body and maturation of their immune system. In literature, it is reported that multiparous sows have the greater gastric capacity and higher feed intakes (Theil et al. 2012); sows are able to convert more energy into milk production compared to the primiparous sow, who is still partitioning energy into her own maturation and growth (Balzani et al 2016; Pluske et al., 1998). Furthermore, sows produce more milk due to having a larger udder (King et al 2002). In conclusion, of course, growth rate depends on time but we can not neglect that gilts are different from multiparous sows.

Theil, P.; Nielsen, M.; Sørensen, M.; Lauridsen, C. Lactation, milk and suckling. In Nutritional Physiology ofPigs; Bach Knudsen, K.E., Kjeldsen, N.J., Poulsen, H.D., Jensen, B.B., Eds.; Danish Pig Research Centre: Copenhagen, Denmark, 2012; pp. 1–47

Balzani, A.; Cordell, H.J.; Sutcliffe, E.; Edwards, S.A. Sources of variation in udder morphology of sows. J. Anim. Sci. 2016, 94, 394–400.

Pluske, J.R.; Williams, I.H.; Zak, L.J.; Clowes, E.J.; Cegielski, A.C.; Aherne, F.X. Feeding lactating primiparous sows to establish three divergent metabolic states: III. Milk production and pig growth. J. Anim. Sci. 1998, 76, 1165–1171.

King, R.H. Factors that influence milk production in well-fed sows. J. Anim. Sci. 2000, 78, 19–25.

6. L 77 fat? Lipid

Authors: Thank you for the correction. We revised the manuscript.

7. L 77 – saturated C15:0 or C16:0? MUFA or PUFA or OCFA

Authors: Thanks for the comment, indeed the parity affect all the main SCFA category including total SFA, total UFA, total MUFA, and total PUFA besides single important FA. For further information you may be interested in the present article: Luise, D.; Cardenia, V.; Zappaterra, M.; Motta, V.; Bosi, P.; Rodriguez-Estrada, M.T.; Trevisi, P. Evaluation of Breed and Parity Order Effects on the Lipid Composition of Porcine Colostrum. J. Agric. Food Chem. 2018, 66, 12911–12920, doi:10.1021/acs.jafc.8b03097.

8. L 77 what are these small metabolites? What do you mention about?

Authors: Thank you for the comment. We were referring to small metabolites that can be detected using the metabolomics technique. Usually are metabolites smaller than 10‐kDa. We revised the text including more information.

9. L 113-114- energy

Authors: We included additional information regarding the diet including its energy.

10. Feeding description for two experiments should be the same.

Authors: Thanks for the suggestion, we commensurated the description.

11. part of 2.3 the description of ELISA protocol is not needed. The detection limit and validation should be given

Authors: Thanks for the suggestion; we removed the description of the ELISA protocol and provided the additional information requested.

12. L 263- it is not discovered for the first time

Authors: we know that this is already known, but it confirms that our study in is line with the literature and therefore it reinforces the other results obtained from the same experiment.

13. L 370 – FA belong to fat

Authors: Thank you for the correction.

14. L 379 – please explain why they have higher concentration of IgA

Authors: Thanks for the suggestion. We revised that part.

15. L 399 – what is health status of mammary gland?

Authors: There is no clear definition of the health status of the mammary gland for sows, it is an adjective that is come commonly used for ruminants. It is based on SCC, electrical conductivity and the presence of mastitis. We revised with sanitary status.  

16. in general the presence of SCC in the milk is not good. Mainly when these cells are leucocytes.

Authors: Thanks for the comment, we agree with the reviewer indeed we mentioned that in the manuscript. We however noted that SCC was positively associated with fat as previously observed by Maurer et al. 2020 and with some piglets performance. Literature for SCC is scarce in sows therefore it is not easy to explain. We included some information based on what is available in the literature.

Round 2

Reviewer 1 Report

The author has gone through the points and changed / revised where needed..

The revised version is acceptable.